# Intelligent Dance Motion Evaluation: An Evaluation Method Based on Keyframe Acquisition According to Musical Beat Features

**DOI:** 10.3390/s24196278

**Published:** 2024-09-28

**Authors:** Hengzi Li, Xingli Huang

**Affiliations:** 1School of Music, Wenzhou University, Wenzhou 325035, China; hengzi@wzu.edu.cn; 2College of Computer Science and Artificial Intelligence, Wenzhou University, Wenzhou 325035, China

**Keywords:** dance movement evaluation, musical beats, keyframe extraction, pose similarity, automated evaluation method

## Abstract

Motion perception is crucial in competitive sports like dance, basketball, and diving. However, evaluations in these sports heavily rely on professionals, posing two main challenges: subjective assessments are uncertain and can be influenced by experience, making it hard to guarantee timeliness and accuracy, and increasing labor costs with multi-expert voting. While video analysis methods have alleviated some pressure, challenges remain in extracting key points/frames from videos and constructing a suitable, quantifiable evaluation method that aligns with the static–dynamic nature of movements for accurate assessment. Therefore, this study proposes an innovative intelligent evaluation method aimed at enhancing the accuracy and processing speed of complex video analysis tasks. Firstly, by constructing a keyframe extraction method based on musical beat detection, coupled with prior knowledge, the beat detection is optimized through a perceptually weighted window to accurately extract keyframes that are highly correlated with dance movement changes. Secondly, OpenPose is employed to detect human joint points in the keyframes, quantifying human movements into a series of numerically expressed nodes and their relationships (i.e., pose descriptions). Combined with the positions of keyframes in the time sequence, a standard pose description sequence is formed, serving as the foundational data for subsequent quantitative evaluations. Lastly, an Action Sequence Evaluation method (ASCS) is established based on all action features within a single action frame to precisely assess the overall performance of individual actions. Furthermore, drawing inspiration from the Rouge-L evaluation method in natural language processing, a Similarity Measure Approach based on Contextual Relationships (SMACR) is constructed, focusing on evaluating the coherence of actions. By integrating ASCS and SMACR, a comprehensive evaluation of dancers is conducted from both the static and dynamic dimensions. During the method validation phase, the research team judiciously selected 12 representative samples from the popular dance game *Just Dance*, meticulously classifying them according to the complexity of dance moves and physical exertion levels. The experimental results demonstrate the outstanding performance of the constructed automated evaluation method. Specifically, this method not only achieves the precise assessments of dance movements at the individual keyframe level but also significantly enhances the evaluation of action coherence and completeness through the innovative SMACR. Across all 12 test samples, the method accurately selects 2 to 5 keyframes per second from the videos, reducing the computational load to 4.1–10.3% compared to traditional full-frame matching methods, while the overall evaluation accuracy only slightly decreases by 3%, fully demonstrating the method’s combination of efficiency and precision. Through precise musical beat alignment, efficient keyframe extraction, and the introduction of intelligent dance motion analysis technology, this study significantly improves upon the subjectivity and inefficiency of traditional manual evaluations, enhancing the scientificity and accuracy of assessments. It provides robust tool support for fields such as dance education and competition evaluations, showcasing broad application prospects.

## 1. Introduction

With the rapid development of information technology, multimedia data analysis based on video and audio has demonstrated immense potential and practical value in various fields. Notably, dance motion analysis, which involves extracting, understanding, and interpreting complex semantic information from extensive dynamic visual data to achieve accurate comprehension, recognition, and evaluation of dance poses, represents a significant challenge [1]. Dance, as a complex and expressive art form, encompasses continuous image frames that cover bodily movements and morphological changes, making it an important application for advanced semantic analysis of images in dynamic scenes and behavior recognition [2]. However, traditional methods of dance motion analysis often face numerous obstacles. Firstly, the redundancy and irrelevance of most motion frames in videos lead to increased computational complexity and can potentially compromise recognition accuracy [1]. Secondly, dance is a time-varying process that requires not only the accuracy of static poses but also the continuity of poses across frames, further compounding the difficulty of analysis. Additionally, current evaluations of dance performances primarily rely on manual expert assessments, which are time consuming, labor intensive, and challenging in terms of the establishment of objective and quantifiable evaluation criteria [3,4]. In contrast, in motion analysis—particularly in basketball shooting—significant progress has been made in quantifying and analyzing joint kinematics using various technologies, including optical motion capture systems (OMCSs) and magnetic inertial measurement units (MIMUs). Research has indicated that these systems can provide detailed and accurate data on arm and wrist kinematics during shooting motions [5,6,7,8]. Notably, the development of hybrid systems combining OMCSs and MIMUs has addressed occlusion issues in marker-based tracking processes, ensuring accuracy and robustness [9]. However, such methods are more inclined towards laboratory-level research, and these sensors may have varying degrees of impact on athletes, leading to a preference for using non-contact sensors such as video and laser sensors for research and evaluation in practical applications, which still poses challenges for analysis methods. Inspired by these advancements in the field of motion analysis, this study proposes a novel method for dance pose estimation and classification that incorporates background music information to enhance the analysis of dance motions. Leveraging the inherent rhythm and meter in dance music, we aim to identify keyframes corresponding to significant motion changes, thereby reducing computational burden and improving analysis accuracy. Our method involves combining music beat detection with pose estimation techniques, such as OpenPose, to capture the human skeletal structure and key points in these keyframes.

A beat extraction method considering prior experience is designed. This method first preprocesses the input sound signal and unifies the sampling rate. Then, it extracts features by calculating the short-time Fourier transform (STFT) and mapping it to the Mel frequency scale. Next, it calculates the energy difference in each Mel band and generates the initial intensity envelope, which is then filtered and smoothed to reduce noise. After that, peaks in the envelope are detected as beat points and normalized. Finally, utilizing the prior knowledge of human perception of a tempo of approximately 120 BPM, it optimizes beat detection through a perception-weighted window, thus enhancing accuracy.A key action frame extraction and feature quantization method based on beats is designed. Leveraging the high correlation between musical beats and dance movements, the problem of manual keyframe extraction and labeling is transformed into a musical beat extraction problem. Image frames corresponding to each beat’s time sequence node are extracted from dance videos and treated as key action frames. This transforms the continuous action recognition and evaluation problem into a discrete action evaluation problem, effectively reducing the computational complexity. Subsequently, key action features are extracted and standardized using the OpenPOSE-based pose recognition method.An intelligent action evaluation method is designed. First, an Action Sequence Evaluation method (ASCS) is constructed based on all action features in a single action (frame) to achieve an accurate evaluation of the overall single action. Then, incorporating contextual features and drawing inspiration from the Rouge-L evaluation method, a similarity measure for action context and rhythm (SMACR) is constructed, focusing on the evaluation of action coherence. Combining the ASCS and SMACR, athletes are comprehensively evaluated from both the static and dynamic perspectives.

## 2. Related Work

Human action recognition (HAR) aims to automatically inspect and identify the nature of actions in unknown video sequences. Due to the growing demand for the automatic interpretation of human behavior, HAR has garnered attention in both academic and industrial circles. In fact, analyzing and understanding human behavior is crucial for a wide range of applications, including video indexing, biometrics, surveillance, and security. The development process of HAR can be summarized as movement vision, action vision, and activity vision [10]. So far, human action is in the research stage of action vision (Action Recognize), which involves extracting certain features from training data, using supervised or unsupervised methods to train a classification model, extracting features from new data and inputting them into the model to obtain classification results [11].

Although small wearable sensors such as [12,13] can also provide information about human movements, they have some limitations, mainly including the potential impact on user comfort, restrictions on data collection (e.g., reliance on user cooperation and sensor placement), difficulty in capturing comprehensive contextual information, poor scalability for different users and scenarios, and challenges in long-term monitoring and real-time data processing [14]. These limitations make wearable sensors less flexible and efficient than machine vision methods in some application scenarios [15,16]. Due to the richness and practicality of images and videos in HAR, visual sensors, especially cameras, have attracted the attention of most researchers. Video action recognition involves processing and analyzing raw graphic or image sequence data collected by sensors using computers to learn and understand human actions and behaviors. It typically involves detecting motion and extracting features, analyzing to obtain human motion patterns, and establishing a mapping relationship between video content and action type descriptions to enable computers to understand videos [17].

Dance motion analysis is an important branch of HAR. Dance motions can be regarded as a series of complex human movement patterns that unfold over time and encompass specific postures, rhythms, and sequences of movements. Machine vision systems understand and recognize dance motions by capturing and analyzing these dynamic changes. Dance motion analysis involves two key tasks: segmenting dance video frame sequences and recognizing dance motions [18]. The segmentation of video sequences is a fundamental task in dance movement analysis based on videos, requiring the identification of the starting and ending frames of an action within a video to represent the maximum amount of data information with the shortest possible behavior sequence [19]. In earlier stages, video frame sequence segmentation employed simple methods to achieve good low-level segmentation of movements. For example, Fod et al. used angular velocity detection of zero-crossing points to segment video motion sequences; Li et al. [20] represented low-level segments (primitives) with the output of a linear dynamic system; and other researchers indicated that motion segmentation was implicitly represented in state machines or motion graphs. However, low-level segmentation only produces incomplete fragments that cannot represent a complete action sequence. In this practical situation, action recognition research has evolved from simple to complex. As a result, current video segmentation requires complex processing according to action categories, such as segmenting results into walking, running, sitting, and so on, which constitute a behavior sequence.

Constructing a hidden Markov model (HMM) based on minimum entropy is a typical unsupervised modeling and segmentation method that maps high-level behaviors to HMM states [21]. This approach involves clustering based on various temporal-scale distance metrics and online segmentation with information loss. Aoki [19] proposed three methods of automatically segmenting long sequences into subaction sequences. The first two methods are applicable to practical videos, meaning that the algorithm iterates through all frames from start to end, creating segmentation whenever breakpoints are encountered. A review of continuous video frame sequence segmentation [19] briefly introduced some researchers’ approaches that equated video frame sequence segmentation to the analysis of video frame sequences, including discrete analysis, point-set analysis, uniform sampling, and temporal-domain modeling. Discrete analysis may consider a particular feature in a behavior sequence but ignores the temporal patterns of human movements. Point-set analysis also disregards the temporal relationship between frames and only focuses on the distribution of data values forming point sets in the sequence [22]. Uniform sampling considers the temporal relationship between frames but assumes that sequences are of equal length, and uses distance functions to measure the similarity between sequences, which is often not the case in real scenarios, where most action sequences are not of equal length. Temporal-domain modeling, on the other hand, removes the limitation on sequence length through modeling frame sequences in temporal order to capture movement changes and determine segmentation points.

In video-based dance movement recognition, the extraction and construction of features, as well as the identification of movements, are crucial steps that significantly impact the accuracy of action recognition. Feature extraction methods can be broadly classified into four categories based on the type of features extracted: low-level features, pose estimation, and image-based and semantic-based methods. In earlier research, most methods for action recognition relied on extracting low-level features of movements, such as the use of Canny edge detection by Carlsson et al. to extract human body movement shape information to represent the edge details of actions, followed by the achievement of human action recognition by matching similar edges [17]. Bruce et al. [23] changed the traditional approach of separately training and sequentially combining pose estimation and action recognition, proposing a framework that integrated pose estimation and action recognition and improving the accuracy of pose estimation.

Methods based on dynamic features often extract human motion information and spatio-temporal features; for instance, Wang et al. [17], inspired by the dense sampling method in image classification, proposed the use of dense optical flow features to describe video content and the utilization of motion boundary histograms to describe dense optical flow features. However, optical flow calculations are complex, and dense optical flow can lead to excessive dimensionality after feature extraction. Semantic-based methods extract mid-level semantic features that can fully utilize all information in the video scene, such as the descriptive features used by Yao et al. [24], including atomic actions, objects, and poses, and the co-occurrence statistics between these descriptive features are modeled, referring to the co-occurrence relationships as “action primitives”. As each type of feature has its own advantages and limitations, in recent years, there has been a tendency to fuse multiple features to form new feature descriptors for action recognition.

However, in current HAR research, there is a paucity of studies that combine audio for research such as dance movement analysis, which may be attributed to the lack of synchronized audio data in datasets. Nevertheless, some studies have begun to incorporate audio into interesting research, such as robot dance programming combined with audio beats [25] and research on music-driven dance algorithms [26]. Beat tracking, a fundamental and continuous research topic in the field of Music Information Retrieval (MIR), has undergone significant advancements in recent years, particularly with the advent of deep learning techniques. This literature review examines two key publications [27,28], focusing on the technical implementations and methodological advancements in beat tracking. In 2019, Jia et al. [27] presented a comprehensive survey of deep learning-based automatic downbeat tracking, emphasizing the system architecture, feature extraction, deep neural network algorithms, datasets, and evaluation strategies. While their work primarily focuses on downbeat tracking, their methodology and techniques are applicable to beat tracking as well. A notable contribution is their discussion of using the Fast Fourier Transform (FFT) for audio signal processing as a precursor to beat detection. This method involves converting the time-domain audio signal into the frequency domain, allowing for the analysis of spectral components that are crucial for beat detection.

However, advancements in feature representation have further refined beat tracking capabilities. As early as 2015, libraries like librosa [28] utilized Mel-frequency cepstral coefficients (MFCCs) as audio features, derived from the Mel-scale frequency bands of an audio signal. These MFCCs provide a more discriminative representation of the audio signal compared to raw spectral features, enhancing the accuracy of beat tracking systems. The emergence of deep learning, particularly Recurrent Neural Networks (RNNs), has revolutionized beat tracking. In the 2020 study by Chuang and Su [29], deep RNNs, specifically Bidirectional Long Short-Term Memory (BLSTM) networks, were applied to symbolic music data for joint beat and downbeat tracking. This work highlights the potential of RNNs in modeling temporal dependencies within music sequences, a crucial aspect of beat tracking. The proposed models were trained in a multi-task learning manner, demonstrating the benefits of joint optimization for related tasks.

Moreover, the study explores the impact of various RNN architectures on beat tracking performance, including BLSTM with an attention mechanism and Hierarchical Multi-Scale RNN (HM-RNN). The attention mechanism enables the model to focus on relevant parts of the input sequence, potentially improving beat detection accuracy, especially in complex musical compositions. Another significant development in beat tracking is the utilization of large-scale datasets and standardized evaluation metrics. Datasets like MusicNet [30] and the Kinetics family [31] provide abundant labeled music clips for training and evaluating beat tracking models. These resources facilitate the comparison of different approaches and encourage the development of more robust systems.

## 3. Methodology

### 3.1. Introduction and Motivation

The relationship between music and dance has a long history, and their interaction and complementarity add rich layers and depth to artistic expression. Music is not only the soul of dance, but also the source of its vitality and emotions. Music provides the foundation of rhythm and melody for dance. The sense of rhythm in dance often comes from the beats and melody changes in music, enabling dance movements to synchronize with music and form a harmonious and unified whole. The rhythm and melody of music can inspire dancers’ body language, guiding them to express the essence of dance in a more expressive way [32]. Most elegant dances are accompanied by beautiful music; therefore, we have reason to believe that there is a high correlation between the design of dance and the rhythm of music, especially in the key movements.

### 3.2. Overall Framework

Based on this correlation, we designed a framework for dance posture recognition and automated evaluation methods based on music beats, as shown in Figure 1. It mainly consists of two parts: preprocessing and intelligent evaluation. (1) First, the beat sequence is obtained by performing beat tracking on the audio in the training data. (2) Then, frames containing dance movements are selected from the dance video based on the beat sequence to construct a keyframe sequence. (3) Pose detection is performed on the movements in the keyframes to obtain quantified standard movement features, thereby forming a “Standard Posture Description Sequence”, which is the key to the entire preprocessing process. The sample to be evaluated also undergoes the same preprocessing steps to obtain the “Pose Sequence to be Evaluated”. (4) Finally, the movement sequence to be evaluated and the standard sequence are evaluated using the SMACR method to assess the accuracy and coherence of the dance movements.

### 3.3. Beat Detection Model

Humans tend to perceive beats in a way that smooths out the intervals between them, meaning that we are more sensitive to the intervals between beats than to their strength. The goal of a beat tracker is to generate a series of beat time points that correspond to the perceived onset points in the audio signal while also forming a regular rhythmic pattern [33]. The onset point of each beat can be described using an onset envelope (OE) [34]. The OE is a time-series related to beat events [35].

Assuming that $\tau_{p}$ represents the ideal beat interval, $\left\{t_{1},t_{2},\ldots ,t_{n}\right\}$ denotes the *n* beat instants that we find, and $O\left(t_{i}\right)$ corresponds to the OE of the *i*-th beat instant. Then, the beat interval between two adjacent beats can be expressed as $\Delta t=t_{i}-t_{i-1}$ [34]. In this context, we define $F\left(\Delta t,\tau_{p}\right)$ to represent the degree of difference between the ideal beat interval and the beat intervals that we find. In this study, we use the square error function applied to the logarithmic ratio of the actual and ideal time intervals, which is given by
(1)$$F\left(\Delta t,\tau_{p}\right)=-{\left(\log\frac{\Delta t}{\tau }\right)}^{2}$$

If the found beats align with the ideal beats (i.e., $\Delta t=\tau_{p}$), then $\Delta t=\tau_{p}$; as the difference increases, $F\left(\Delta t,\tau_{p}\right)$ gradually becomes negative.

In this way, we can construct an objective function $C\left(\left\{t_{i}\right\}\right)$ related to $O\left(t_{i}\right)$ and $F\left(\Delta t,\tau_{p}\right)$, which aims to maximize OE while minimizing $F\left(\Delta t,\tau_{p}\right)$:(2)$$C\left(\left\{t_{i}\right\}\right)=\sum_{i=1}^{n}O\left(t_{i}\right)+\omega \sum_{i=2}^{n}F\left(\Delta t=t_{i}-t_{i-1},\tau_{p}\right)$$
where $\sum_{i=1}^{n}O\left(t_{i}\right)$ represents the maximum intensity of all OEs, $F\left(\Delta t,\tau_{p}\right)$ evaluates the difference between two adjacent beats and the ideal beat $\tau_{p}$, and ω is a weight value obtained through learning, which is used to balance the importance of the OE and *F*.

The key of the objective function lies in obtaining the combination of $O\left(t_{i}\right)$ and $F\left(\Delta t,\tau_{p}\right)$–namely, $O^{*}\left(t\right)$—that is as large as possible:(3)$$C^{*}\left(t\right)=O\left(t\right)+\max_{\tau =0\ldots t}\left\{\omega F\left(t-\tau ,\tau_{p}\right)+C^{*}\left(\tau \right)\right\}$$

Here, $O\left(t\right)$ represents the local onset strength at time *t*, which is also the highest score at that moment. It is combined with the highest score from the previous beat time $C^{*}\left(\tau \right)$ that maximizes the sum of this highest score and the transition cost starting from that moment. During the calculation of $C^{*}$, we also record the previous beat time $P^{*}\left(t\right)$ that actually provides the highest score:(4)$$P^{*}\left(t\right)=\underset{\tau =0\ldots t}{\arg\max}\left\{\omega F\left(t-\tau ,\tau_{p}\right)+C^{*}\left(\tau \right)\right\}$$

Due to the limitation of the beat error $F\left(\Delta t,\tau_{p}\right)$, the error cannot exceed one beat, meaning that we only need to search within a limited range of τ. Since the optimal beat time is unlikely to deviate significantly from the $t-\tau_{p}$ position, we can set the search range to be around $t-2\tau_{p}\ldots t-\tau_{p}/2$ as shown in Figure 2.

### 3.4. Onset Strength Envelope Calculation

Similarly to many other onset models [36], we compute the onset envelope from a simple perceptual model. Firstly, the input audio is resampled to 8 kHz. Then, the spectrogram of the short-term Fourier transform (STFT) is calculated with a typical window size of 32 ms and a step size of 4 ms.

Next, the STFT magnitudes (spectrogram) are computed using a 32 ms window and a 4 ms step size between frames. These are then converted into an approximate auditory representation by mapping to 40 Mel bands [37]. The Mel spectrogram is converted into dB units, and the first-order difference over time is computed in each band. Negative values are set to 0 through half-wave rectification, and the remaining positive differences across all frequency bands are summed. This signal is passed through a high-pass filter with a cutoff frequency of approximately 0.4 Hz to make its local mean zero and smoothed through convolution with a Gaussian envelope with a width of approximately 20 ms. This results in a one-dimensional onset strength envelope that varies over time and responds to proportional increases in energy across approximate auditory frequency bands. Figure 3, Figure 4 and Figure 5 show examples of the STFT spectrogram, Mel spectrogram, and onset strength envelope for the first 10 s of Sample 1. Peaks in the onset envelope clearly correspond to moments of significant energy onset across multiple frequency bands in the signal. As the balance between the two terms in the objective function in Equation (Equation 2) depends on the overall scale of the onset function, which may itself depend on the characteristics of the instrument playing or other signal spectral aspects, we normalize the onset envelope for each music excerpt by dividing it by its standard deviation.

### 3.5. Global Tempo Estimation

As Equation (Equation 2) relies on prior knowledge of the ideal beat interval, there is a significant correlation between the onset strength envelope and the beats. Specifically, for a periodic signal, large correlations will also appear at any integer multiples of the fundamental period. While we may not be able to select the maximum peak that perfectly aligns with the beats, it is possible to select a secondary (local) peak to match, and human perception of rhythm tends to favor approximately 120 BPM [37], which is prior knowledge. Therefore, a perceptually weighted window can be applied to the raw autocorrelation to reduce the weight of periodic peaks that deviate significantly from this preference. The scaled peaks can then be interpreted as indicators of the likelihood that humans would select that period as the basic tempo. In this way, the tempo period strength (TPS) can be expressed as follows:(5)$$TPS\left(\tau \right)=W\left(\tau \right)\sum_{t}O\left(t\right)O\left(t-\tau \right)$$
where $W\left(\tau \right)$ is a Gaussian weighting function of the logarithmic time axis quotient:(6)$$W\left(\tau \right)=\exp\left\{-\frac{1}{2}{\left(\frac{\log_{2}\tau /\tau_{0}}{\sigma \_\tau }\right)}^{2}\right\}$$
where $\tau_{0}$ is the center of the tempo period preference, and the parameter $sigma_{\tau }$ controls the width of the weighting curve (expressed in octaves due to the use of $\log_{2}$). The primary tempo period estimate is simply the value of τ that maximizes TPS(τ). To obtain $\tau_{0}$ and $\sigma_{\tau }$, we directly use the trained results from [38].

### 3.6. Keyframe Extraction

When we obtain the music beat and the beat interval $\tau_{p}$, we can derive the beat time-series $T=\left\{t_{1},t_{2},\ldots ,t_{n}\right\}$ as shown in Figure 6. The frame $f_{i}$ extracted from the same position in the dance video based on time point $t_{i}\in T$ in this series constitutes a keyframe, and the corresponding sequence of these keyframes is the keyframe sequence $F=\left\{f_{1},f_{2},\ldots ,f_{n}\right\}$.

## 4. Intelligent Dance Motion Evaluation

### 4.1. Dance Pose Recognition

We introduce a deep learning-based method for standardized evaluation and human pose estimation in [39]. This approach first utilizes OpenPose for body joint detection, followed by a deep neural network (DNN)-guided pose information extraction strategy. Finally, leveraging the team’s extensive experience in dance teaching, a novel method is proposed to describe and distinguish differences in dance moves. This method enables quantitative evaluation and provides intuitive feedback on dance movement mechanisms, thereby enhancing the monitoring of participants’ progress. Specifically, the method consists of the following parts: (1) skeleton and key point extraction; (2) pose representation; (3) DNN-regression-based pose estimation; (4) cascaded pose regression; (5) feature description and determination of dance movement; and (6) pose matching of human movement. In this study, we apply this method to recognize human poses in keyframes and obtain the feature descriptions of skeleton and key points, with which a basic scoring method is constructed.

ST-AMCNN [40], similar to the method proposed in this paper, also chooses to extract posture information from the original images using the OpenPose algorithm, with a primary focus on the upper arms, forearms, thighs, calves, and trunk. Subsequently, it employs a Spatial Transformer Network (STN) for pose alignment. The input learner’s pose image is aligned with the standard pose image through STN to optimize the transformation matrix and reduce pose discrepancies. Finally, the Attention-based Multi-Scale Convolution (AMC) module, which is based on a twin network architecture, is employed for feature extraction and matching. This module is capable of extracting multi-scale features while focusing on useful pose features and ignoring irrelevant features such as backgrounds. The ST-AMCNN model achieved the highest AP value (98.15%) in the overall pose matching task. For incorrect pose pairs, the matching score of the ST-AMCNN model (0.7077) is significantly higher than that of other compared models, indicating its strong ability to distinguish between incorrect poses. This led us to choose it as a comparison method.

### 4.2. Basic Scoring

In [39], we leveraged the similarity of key movements, such as limb angles, to grade—and, thereby, assign a score to—a keyframe (or an action frame to be evaluated). A standard action in the reference frame requires a $90^{\circ }$ angle, while the actual non-standard action deviates from this standard by an error of delta theta [40]. To reasonably describe the magnitude of this deviation, this study uses the cosine similarity between two sequences to represent the degree of similarity between the action being evaluated and the standard action.

Given a sequence of *n* action frames to be evaluated, denoted as $F=\{f_{1},f_{2},\ldots ,f_{n}\}$, we perform a similarity evaluation for each frame and obtain a score sequence $S=\{s_{1},s_{2},\ldots ,s_{n}\}$. Clearly, for the movements of the tested subject, a higher score *S* is preferred. Furthermore, we can use the mean value of *S*, denoted as $\bar{S}$, as the overall score to evaluate the current test results:(7)$$\bar{S}=\frac{\sum_{i=1}^{n}s_{i}}{n}$$

Obviously, this evaluation method is quite simple, merely involving the simple averaging of similarity evaluations (scores) between each action feature and the standard action feature, without considering the coordination among multiple action features that constitute a complete action. Therefore, we can consider multiple action features in a keyframe as a set to be evaluated and compare this evaluation set with the corresponding standard action feature set holistically. For a sequence of action frames *F*, where each sequence has *m* evaluable action features, let $\theta_{i}^{j}$ represent the *j*-th evaluable feature of the *i*-th frame ($j\in \left[1,m\right]$). Then, we can obtain *m* evaluable feature sequences $Q^{j}=\left\{\theta_{i}^{j},\theta_{i}^{j},\ldots ,\theta_{n}^{j}\right\},j\in \left[1,m\right]$, which are comparable to the reference feature sequence $\Theta^{j}$. In this study, the cosine similarity between the two sequences is used to represent the degree of similarity between the evaluated action and the standard action:(8)$$s^{j}=\frac{Q^{j}\Theta^{j}}{∥Q^{j}∥\times ∥\Theta^{j}∥}=\frac{\sum_{i=1}^{n}Q_{i}^{j}\times \Theta_{i}^{j}}{\sqrt{\sum_{i=1}^{n}{\left(Q_{i}^{j}\right)}^{2}\times \sqrt{\sum_{i=1}^{n}{\left(\Theta_{i}^{j}\right)}^{2}}}}$$
where $s^{j}$ represents the score of the *j*-th action. Therefore, the overall score of the entire action can be denoted as $\bar{S}$, and it can be referred to as the “average score based on the cosine similarity of action sequences (ASCS)”.

### 4.3. Scoring Method Based on Action Contextual Relationships

It is evident that this scoring method only considers the accuracy of a single frame’s movement without taking into account the contextual relationship between the current frame and the current action. However, the coherence of dance movements is often more crucial than the accuracy of a single movement. For example, it is worth noting that the *j*-th evaluable feature may deviate significantly from the standard action. Therefore, the dancer may consciously adjust the action within a short period of time following this deviation. However, this process may be too short to be reflected in adjacent keyframes. To address this, we draw inspiration from the Rouge evaluation method in natural language processing (NLP) [41] and adopt the more stringent Rouge-L (Rouge-longest common subsequence, Rouge-LCS) metric to evaluate the dancer’s performance, resulting in the evaluation method. Based on the evaluation of angle similarity in Section 3.6 of [20], the Scoring Method Based on Action Contextual Relationships (SMACR) is proposed:(9)$$R_{LCS}=P_{LCS}=\frac{LCS\left(Q^{j},\Theta^{j}\right)}{n}$$
(10)$$F_{LCS}=\frac{\left(1+\beta^{2}\right)R_{LCS}P_{LCS}}{R_{LCS}+\beta^{2}P_{LCS}}$$

Here, as the lengths of $\Theta^{j}$ and $Q^{j}$ are both *n*, we have $R_{LCS}=P_{LCS}$, as well as $F_{LCS}=R_{LCS}$, which means that the evaluation is performed directly using Equation (Equation 8). Considering that Equation (Equation 8) uses the percentage of similarity as an evaluation metric, we set a threshold filter here: for example, filter = 0.85, which represents the acceptable level of action similarity during manual evaluation. This allows the original similarity score to be converted into a 0–1 sequence for comparison with the standard action. Figure 7 demonstrates the calculation process of SMACR. It can be observed that the average similarity score for a single keyframe is 86.1%, which is significantly higher than the score of 55.6% calculated using SMACR. The average score evaluates the accuracy of individual movements, while the SMACR score assesses the continuity of movements. The quality of individual movements is positively correlated with the continuity score. In dance training, it may be advisable to first focus on the accuracy of individual movements and then gradually consider the continuity of movements. This allows us to evaluate the quality of dance movements from two aspects simultaneously, which is also the key advantage of our method over other automated evaluation methods.

## 5. Experimental Results and Discussion

### 5.1. Dataset

For the dataset, we select the popular dance game *Just Dance* [42]. Although this “game” is not as rigorous and standardized as professional dance, it boasts immense influence, diverse dance styles, and a wide audience. It also comes with a scoring system that we can use as a reference for evaluation. In our experiments, we select ten popular tracks, and their key parameters are shown in Table 1. The difficulty level is divided into four grades: 1 = children’s version, 2 = easy, 3 = moderate, and 4 = difficult. In the original samples, there is an indicator named “sweat difficulty”, which reflects the intensity of the dance music to a certain degree: the greater the movement amplitude, the more sweat produced. Therefore, in the analysis, we need to take into account not only the basic difficulty (the difficulty of the movements) but also this “sweat difficulty”.

It is worth noting that the original samples were divided into two directories: a preview directory and a complete file directory. Taking the “BeNice” track used in the example analysis as an instance, the preview directory only contained a 30-s clip, while the full track was 160 s long. In addition to the length, the resolution of the corresponding video files was also changed, which had a certain impact on the recognition of OpenPOS.

### 5.2. Music Beat Recognition

We developed a framework for music beat recognition based on the librosa library in Python 3.10.11 [43], optimizing the parameters of the original library for our purpose. Firstly, we utilized librosa.beat.beat_track for beat detection, as it estimated the beat locations by analyzing the onset envelope and spectral features of the audio signal. This function returned an estimated beat tempo (expressed in BPM, beats per minute) and the frame indices corresponding to each beat. The detection results are shown in Figure 8. It can be observed that a beat sequence was detected, but the results were not ideal, with multiple unequal intervals present.

Based on the beat detection method described in Section 2, we first calculated the log-power features (Figure 9a) and then obtained the power spectrum features (Figure 9b) before conducting beat detection. The detected beats are shown in Figure 10. It can be observed that the beats detected using the method in Section 2 exhibited strong regularity and high alignment with the actual beats, significantly outperforming the beat detection method based on spectral features.

The beat detection results for all 12 samples are shown in Figure 11. Among them, Sample 10 had the highest onset strength, so we used the legend settings of Sample 10 to format the beat detection results for the other samples. It can be seen that the beats of Samples 1, 3, 5, 7, 8, and 10 were relatively strong, allowing us to obtain shorter intervals and more key points. Comparatively, Samples 2, 5, 9, 11, and 12 had more gentle beats with longer intervals, and the secondary beats were ignored.

Sample 5 was very special. One can see that there was no sequence from the 10th to the 15th second, as this track had a significant silent period during this time as shown in Figure 12. The current beat detection algorithm was not able to detect beats in this interval, but the dance performance continued. To handle this special case, there were two strategies to choose from: (1) directly ignoring this interval because there were no beats and, therefore, no corresponding keyframes; or (2) applying the same keyframe extraction method to this interval using standard intervals as TPS considered global beats. The former strategy was chosen for this study, as it could simplify the preprocessing flow. Moreover, as this part accounted for a very low proportion of the entire sample, it would not have a significant impact on the final evaluation results.

### 5.3. Dance Keyframe Evaluation Based on Musical Beats

Based on the beats obtained as shown in Figure 13, we selected corresponding keyframes from the original samples to construct the GroundTruth evaluation dataset. Then, we extracted the corresponding frames from the actual dance videos of the dancers to build the Real Results dataset for evaluation. This is shown in Figure 11.

We then utilized the evaluation method described in Section 6 to assess the results. The evaluation results are shown in Table 2. It can be observed that there were significant differences in the scores obtained by the tested individuals when using different evaluation criteria. The *Just Dance* scoring system tends to be relatively lenient, focusing mainly on the continuity of movements, which is similar to the SMACR evaluation. Songs with higher *Just Dance* scores also received higher scores from the SMACR evaluation. However, the SMACR scores were generally lower than the average keyframe scores, reflecting the sensitivity of SMACR to the perception of adjacent movements. In comparison, although the testers achieved high key scores at different levels of difficulty, indicating that the testers’ accuracy in key movements (similarity to standard movements) was relatively high, the overall SMACR scores showed a negative correlation with difficulty, suggesting that movements with higher difficulty, due to their greater range of motion and faster pace, resulted in lower SMACR scores, which focused on the continuity of movements.

Compared to using all video frames for action matching tests, selecting keyframes based on musical beats significantly reduced the computational load for matching. Among the 12 test samples, an average of 2–5 keyframes were selected every two seconds (corresponding to 2–5 beats), which is much lower than the 24 frames per second of the full frames, reducing the computational load to 4.1–10.3% of the original (as shown in Table 2, in other words, it was accelerated by 9.2 to 15.25 times as shown in Table 3). It is noteworthy that when evaluating using only these keyframes, the overall evaluation accuracy decreased by only 3%. Secondly, in terms of matching accuracy for individual action frames, ST-AMCNN outperforms the ASCS method proposed in this paper. This is mainly because although the angle-based cosine matching method adopted by ASCS is simple, its matching accuracy significantly decreases when there are significant differences between the test subject and the standard sample, and the accuracy of multi-scale matching is also not high. However, the advantage of the ASCS method lies in its lack of need for pretraining and its smaller computational load. Lastly, both ST-AMCNN and the ASCS method proposed in this paper can shift from evaluating individual action frames to evaluating action continuity with greater emphasis after incorporating the Rouge-L evaluation mechanism. Therefore, we present the fourth set of comparison test results here, namely, ST-AMCNN*. It can be seen that, similar to the test results of SMACR, the improved evaluation method of ASCS, the score for continuity evaluation is lower than that for individual action frames, but the overall score is still higher than that of the SMACR method. In summary, selecting appropriate keyframes helps reduce the computational cost of subsequent matching methods, and depending on the specific application scenario, a faster evaluation method (such as SMACR in this paper) or a method with higher accuracy (such as ST-AMCNN) can be chosen.

Overall, the evaluation of ASCS is more suitable for assessing individual or static movements and is appropriate for learning and training purposes. On the other hand, SMACR considers both the accuracy and fluency of movements, making it more suitable for quick scoring during athletes’ later stages of training and competitions.

## 6. Conclusions

In order to address the issues of strong reliance on manual subjective experience and high evaluation costs in the teaching and assessment of competitive sports such as dance, this study proposes a dance gesture recognition and intelligent evaluation method based on the strong correlation between dance and musical beats. First, audio analysis is used to accurately obtain musical beats. Then, corresponding image frames are extracted from dance videos based on the musical beat sequence to construct a keyframe sequence. Next, posture recognition methods are employed to quantify the human movements in the keyframes into motion features. Finally, the accurate evaluation of individual frames (motions) is achieved through motion feature similarity evaluation methods, and the continuous evaluation of motion sequences is achieved through contextual methods. This enables the evaluation method proposed in this study to provide a more reasonable and accurate assessment of the dance movement accuracy from two perspectives. Tests on public datasets demonstrate the effectiveness of this method.

Although this paper conducted experiments on the publicly available *Just Dance* dataset, there are still some limitations. One notable deficiency is the lack of generalization testing on a wider range of datasets. Of course, this is also related to the nature of the publicly available datasets such as KTH [44], Weizmann [44], etc. Most of these datasets do not provide or include synchronized audio files, which prevents us from establishing an effective beat, further extracting keyframes, and utilizing them for posture and action evaluation.

## Figures and Tables

**Figure 1 sensors-24-06278-f001:**
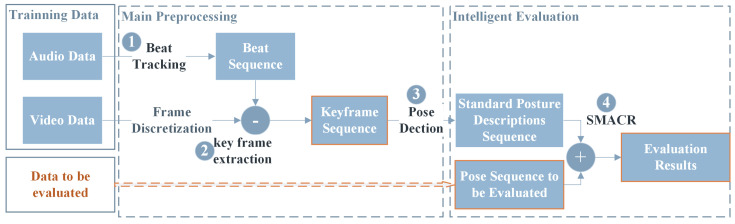
Overall framework.

**Figure 2 sensors-24-06278-f002:**
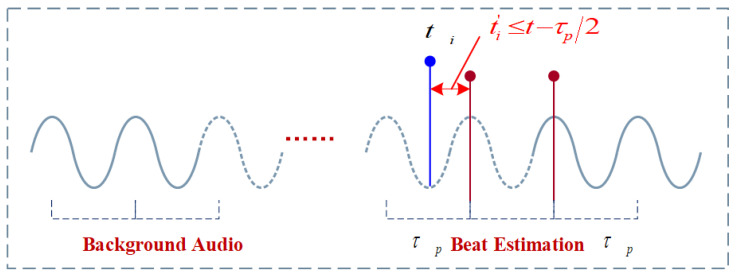
Constraints on the search range in beat detection.

**Figure 3 sensors-24-06278-f003:**
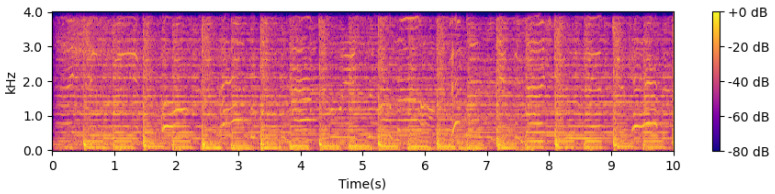
Linear frequency spectrogram.

**Figure 4 sensors-24-06278-f004:**
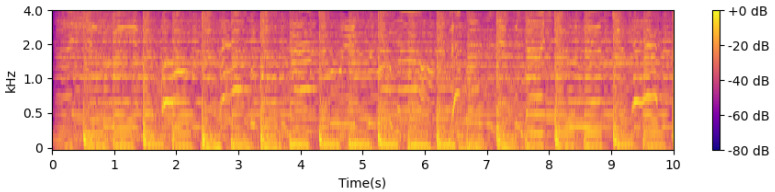
Mel frequency spectrogram.

**Figure 5 sensors-24-06278-f005:**
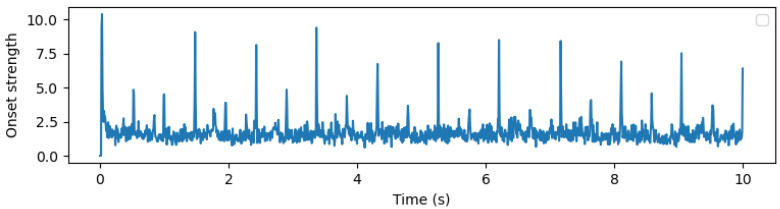
Onset strength envelope.

**Figure 6 sensors-24-06278-f006:**
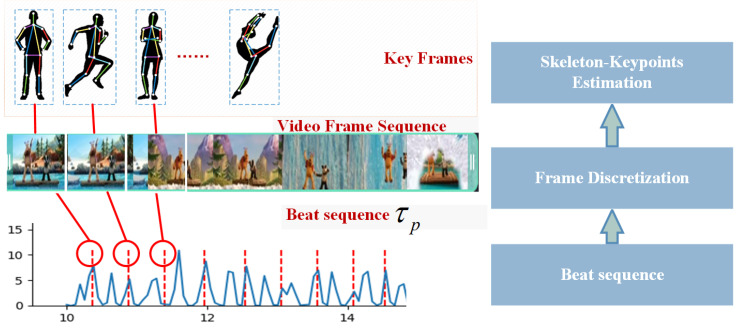
Keyframe extraction.

**Figure 7 sensors-24-06278-f007:**
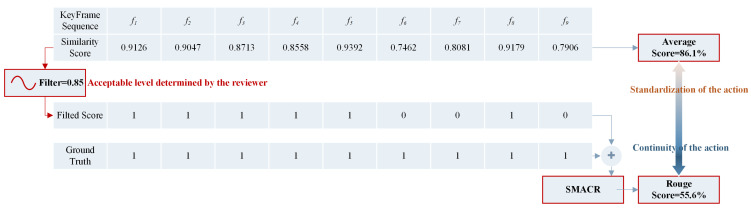
Schematic diagram of the SMACR calculation process.

**Figure 8 sensors-24-06278-f008:**
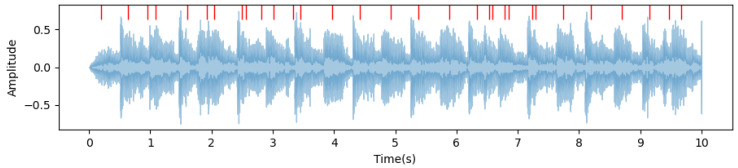
Frequency spectrogram and beat detection results for Sample 1: Partial illustration of the first 10 s.

**Figure 9 sensors-24-06278-f009:**
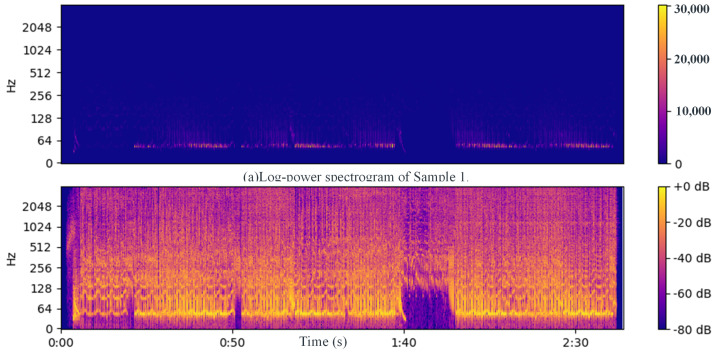
The characteristics of the audio in Sample 1.

**Figure 10 sensors-24-06278-f010:**
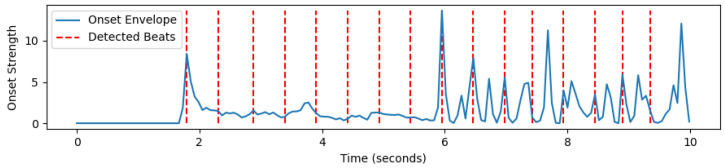
Beat detection results for Sample 1: Partial zoom-in (first 10 s).

**Figure 11 sensors-24-06278-f011:**
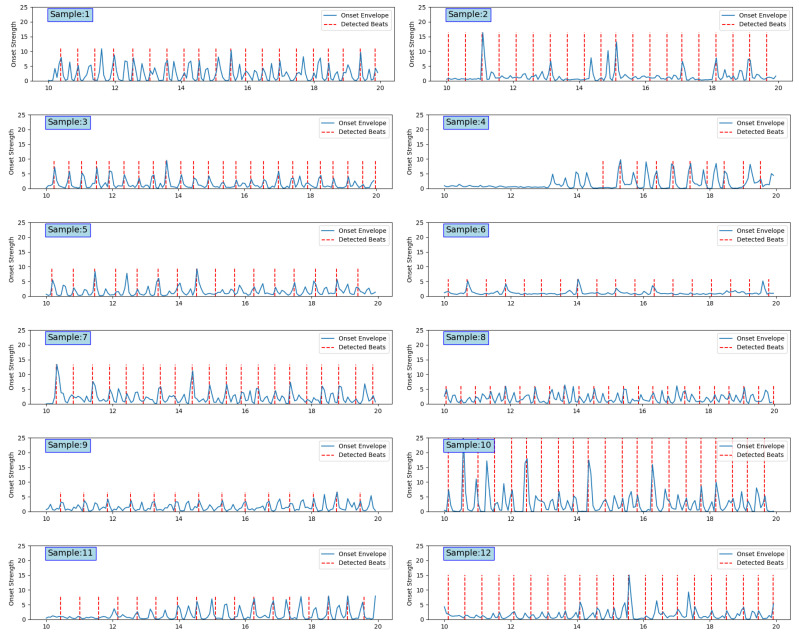
Batch results.

**Figure 12 sensors-24-06278-f012:**
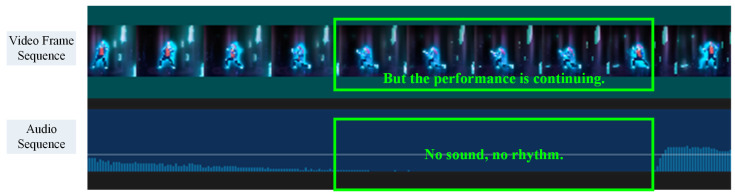
Special case: Sample 4.

**Figure 13 sensors-24-06278-f013:**
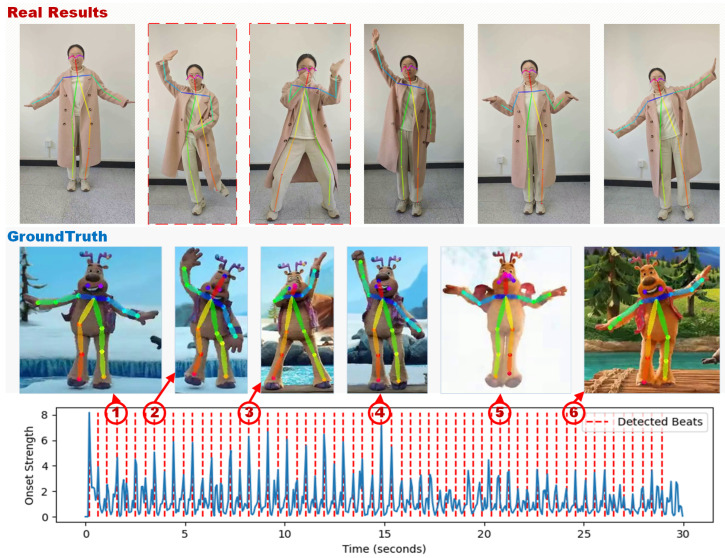
Comparative diagram of the ground truth and actual test sets.

**Table 1 sensors-24-06278-t001:** Sample set.

Sample No.	Title	Artist	Sweat Difficulty	Difficulty
1	AnythingIDo	CLiQ	3	4
2	BoyWithLuvNextALT	BTS	3	4
3	MORE	K/DA	3	4
4	RatherBe	Clean Bandi	3	3
5	BringMeToLife	Evanescence	3	3
6	LoveMeLand	Zara Larsson	2	3
7	Telephone	Lady Gaga	2	2
8	SweetButPsycho	Ava Max	2	2
9	AsItWas	Harry Styles	2	2
10	SissyThatWalk	RuPaul	2	1
11	Magic	Kylie Minogue	1	1
12	BeNice	The Sunlight Shakers	2	1

**Table 2 sensors-24-06278-t002:** Test result.

Sample No.	All Frames				Keyframes			
	**ST-AMCNN**	**ST-AMCNN***	**ASCS**	**SMACR**	**ST-AMCNN**	**ST-AMCNN ***	**ASCS**	**SMACR**
1	96.99	68.87	93.25	63.75	98.77	71.5	94.73	59.16
2	89.63	55.97	90.11	56.3	92.14	59.13	92.69	57.65
3	95.39	56.86	93.39	54.01	97.33	59.71	96.17	57.03
4	88.19	55.16	87.45	55.07	90.16	58.17	89.83	73.36
5	91.84	71.87	91.75	70.33	94.63	75.47	93.62	76.49
6	91.44	73.87	90.65	73.9	93.7	76.98	92.77	59.13
7	95.63	61.01	93.44	56.84	97.41	64.43	94.96	67.78
8	89.69	68.08	90	64.43	92.63	70.08	92.6	69.43
9	94.9	69.53	95.08	66.17	97.52	73.03	96.62	72.57
10	95.83	73.26	92.09	69.04	97.36	76.6	93.97	62.51
11	93.98	63.85	90.96	60.14	96.89	66.59	92.52	71.29
12	93.12	67.87	91.08	69.35	94.62	71.36	92.98	66.3
AVG.	93.05	65.52	91.6	63.28	95.26	68.59	93.62	66.06

* Used a continuous evaluation method similar to ROUGE-L (corresponding to the SMACR method).

**Table 3 sensors-24-06278-t003:** Test result: speed up.

Methods	1	2	3	4	5	6	7	8	9	10	11	12	AVG
ST-AMCNN	10.2	11.5	10	10.4	12.7	12.9	11.4	9.36	13.4	10.3	12.9	11.2	11.3549
ASCS	11.3	11.7	10.8	11.8	14	13.9	12.5	11.1	15.2	12	13.7	12.8	12.5735

## Data Availability

The data source used was part of the game *Just Dance* published by UBISOFT (https://www.ubisoft.com.cn/jdc/ (accessed on 7 September 2023)). Researchers can access relevant data through the game’s local cache files. When using these data, researchers should abide by the relevant statements of UBISOFT. This article only used these data for research purposes and did not involve any commercial activities.

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
