# Peer review of "Intelligent Dance Motion Evaluation: An Evaluation Method Based on Keyframe Acquisition According to Musical Beat Features"

_sensors, 2024, doi:10.3390/s24196278_

Round 1
Reviewer 1 Report
Comments and Suggestions for Authors
General comments
Since the authors have selected an interesting research topic, their work should be acknowledged. However, the manuscript incurs in scientific writing in all sections of the manuscript. For example, the abstract has at least 450 words! An abstract should bring the main essence of the manuscript. At least sixteen lines were spent on the background! In the introduction section, the authors quote only two references! This specific section is a mix between introduction and methods section. It is impossible to follow the proposal of the authors with this current paper. My suggestion is to reject and authors must improve all sections of the manuscript.
Comments on the Quality of English LanguageThe current paper must be revised by an native english speaker. In this current form it is impossible to follow the scientific language.
Author Response
- We have rewritten the entire abstract. Firstly, we significantly streamlined the background introduction to swiftly transition to the issues faced in action detection and evaluation. Subsequently, we present the solution proposed in this paper. We provide a more detailed description of the crucial components, including the preprocessing part consisting of beat detection, key frame extraction, and action recognition, as well as the action accuracy evaluation part.
- We have substantially revised the content of the introduction and organized it into four parts as per the framework: 1) the existing problems with manual evaluation; 2) the need for video analysis methods; 3) the issues with video analysis; 4) the construction of an appropriate analysis method and the main contributions of this paper.
- We have enhanced the related work section according to the new outline framework and research content. We first introduced the field of Human Action Recognition (HAR), then presented the current research status of dance action recognition in dance movements, thereby highlighting video sequence segmentation and action recognition as key research focuses and challenges. We also emphasized the research status and development of these two parts. From the perspective of existing research, music beat analysis appears to be a separate branch of research. This is also reflected in the fact that while existing HAR-related data tends to be multimodal, audio data is not included in related research. In recent years, there have been only a few interesting studies, such as combining audio beats for robot dance programming and research on music-driven dance algorithms. This makes the preprocessing based on music beats in this paper to support dance action analysis somewhat novel.
- We have revised the English writing of the paper as much as possible and then applied for MDPI's English language editing service to improve the language organization level of this paper as much as possible.
- We have rearranged the framework of the paper. The original Chapters 2 and 3 have been merged and enhanced into a new Chapter 2, which describes how to use audio data to calculate and obtain acceptable music beats and how to use music beats to extract key frames from synchronized dance video sequences (which are candidate frames to be evaluated in the test data). The action sequences constituted by the dance actions in these key frames are the basis for subsequent action accuracy and coherence evaluation.
The original Chapters 4 and 5 have been merged into a new Chapter 3. In this chapter, we first introduce in detail the similarity evaluation method for a single dance action (in a single key frame), which is based on the evaluation method of key feature angle differences shown in Figure 7. In this way, all action features in a single key frame can form a feature sequence. Based on this feature sequence, a similarity evaluation method is constructed and compared with a standard feature sequence, thereby enhancing the overall value of action evaluation. This is also one of the important improvements of this paper compared with other methods.
Furthermore, to consider the continuity of actions, we innovatively introduced the Rouge-L evaluation method from NLP, treating the action corresponding to a single key frame as a word/token, and converting the evaluation of the entire action sequence into a similarity evaluation similar to that between sentences. This enables the entire action evaluation algorithm to evaluate from both the accuracy of individual actions and the coherence of actions, making the evaluation results more accurate and reasonable.
- After completing the above revisions, we reexamined the theme of the paper and revised it to "Intelligent Dance Motion Evaluation: An Evaluation Method Based on Key Frame Acquisition According to Musical Beat Features."

Reviewer 2 Report
Comments and Suggestions for Authors
General considerations:
The paper refers to a study that propose a proposes an novel audio-video data alignment method that improves the accuracy of dance movement analysis.
The study makes a good connection between mathematical models, processing algorithms, experimental data processing and the analysis and interpretation of the results
Some comments for authors:
The connection between the definition of the SMACR and SMACR + factors in paragraph 5.2 with the results in table 2 (chapter 6) is not clear enough.
The test results in table 2 are not analyzed and evaluated compared to other sources/works, nor are they explained if and how good they are, the values ​​obtained are sufficiently high or low.
Also, the relationship between the values ​​obtained for SMACR and SMACR + must be analyzed.
The conclusions are too hasty, given the fact that the results are not compared with others obtained by other researchers and thus, they are not concrete enough.
The authors have no works cited in the bibliography. Is this a lack of previous publications or just a shortcoming of the authors?
Other observations
Raw 300: "The detected beats are shown in Figure ?? and Figure 10" to be corrected
Figure 8 is not clear enough
Figure 7 why is it only given for Sample1? Other samples could be described.
Author Response
1. The related work has been enhanced according to the new outline framework and research content. It begins with an introduction to the field of Human Action Recognition (HAR), followed by an overview of the current research status on dance motion recognition in dance movements. This leads to the identification of video sequence segmentation and action recognition as key research focuses and challenges, with a focus on introducing the current research status and developments in these two areas. From the perspective of existing research, music beat analysis appears to be an independent research area. This is also reflected in the fact that although there is a multimodal development trend in existing HAR-related datasets, audio data is not included in related research. In recent years, there have been only a few interesting studies, such as combining audio beats for robot dance programming and research on music-driven dance algorithm, which makes the preprocessing based on music beats conducted in this paper to support dance motion analysis somewhat novel.
- After re-examining and reorganizing the paper framework, we deemed the description of SMACR and SMACR+ unreasonable and thus made significant revisions. In the newly revised Chapter 3, we reorganized the description of this part: Firstly, we introduced in detail the similarity evaluation method for single dance actions (in a single keyframe), which is based on the evaluation method of key feature angle differences shown in Figure 7. In this way, all action features in a single keyframe can form a feature sequence. Based on this feature sequence, a similarity evaluation method is constructed and compared with the standard feature sequence, thereby enhancing the overall value of action evaluation. This is also one of the important improvements compared with other methods, and it is named "average score based on the cosine similarity of action sequences (ASCS)". Furthermore, to consider the continuity of actions, we innovatively introduced the Rouge-L evaluation method from NLP, treating the actions corresponding to a single keyframe as a word/token, and transforming the evaluation of the entire action sequence into a similarity evaluation similar to that between sentences. This enables the entire action evaluation algorithm to evaluate from both the accuracy of individual actions and the coherence of actions, making the evaluation results more accurate and reasonable. This improved method is named "Scoring Method Based on Action Contextual Relationships (SMACR)".
Based on this adjustment, Question 3) can be explained as the difference between ASCS and SMACR, where ASCS describes the accuracy (similarity) of actions in a single keyframe, while SMACR describes the similarity (coherence) between keyframe sequences, i.e., action sequences.
- 4. and 5) There is a series of datasets and research in the field of HAR, which are introduced in the new literature review. However, these methods focus on the accuracy of action detection and action classification, and the datasets mainly consist of videos, infrared, and depth images, with little simultaneously containing audio data corresponding to or synchronized with the videos. Therefore, it is difficult to adapt to the needs of music beat detection preprocessing in this paper. The research in references [10] (Research on the Construction of Music Performance Robot Based on Beat Recognition) and [11] (Research on Music Driven Dance Generation Algorithms) has similarities with this paper, but we were unable to obtain the corresponding public sample data. Existing experiments have mostly proven the effectiveness of this method and a series of optimizations and improvements, so there are limitations in cross-dataset validation. We also pointed this out in the conclusion section.
- We have revised some errors in the paper, including word spelling, literature, and figure application errors. The original Figure 8 has been enhanced; in Figures 13 and 14, we have added descriptions and beat extraction situations for 11 other samples, analyzed the characteristics of different samples in detail, as well as the special situation in Sample 4 and the processing strategy when music beats are missing.
- Furthermore, in consideration of the limitations in our English writing skills, after revising the paper, we chose to use English polishing services to strive for accurately expressing the main viewpoints.

Reviewer 3 Report
Comments and Suggestions for Authors
This paper presents a novel method for aligning and evaluating dance gesture movements with musical beats using multi-sensor data. The approach aims to enhance the accuracy and efficiency of dance movement analysis by leveraging the correlation between music and dance. Here are some suggestions:
1. The title of the article is "Multi-sensor Alignment," but there is little research on alignment in the text. The main focus is on how to identify audio beats and perform fine-grained pose recognition on video frames corresponding to these beats to evaluate dance movements.
2. Some sections of the paper, particularly the methodology, could benefit from clearer explanations and better structure to enhance readability. More visual aids, such as flowcharts or diagrams, could help illustrate the process and make it easier for readers to follow. In addition, the transitions between parts in the calculation method of music beats are abrupt and could use clearer connections and diagrams to guide the reader through the methodology.
3. The paper introduces a similarity scoring method for evaluating dance movements, but it lacks detailed evaluation metrics and comparative analysis with existing methods. Including a table that compares the proposed method's performance with other methods using standard metrics such as precision, recall, and F1-score would offer a more comprehensive evaluation. The current evaluation method is difficult to prove superiority due to the lack of a unified evaluation standard.
4. The writing of the paper needs improvement, as there are some noticeable errors. For example, on page 2, line 65, "esigned" is likely a spelling error. On page 9, line 300, the figure number is "Figure ??". On page 4, line 148, on page 10, lines 305 and 308, the figure numbers Figure 6, Figure 9 and Figure 10 are incorrect.
Author Response
- We have made significant adjustments to the overall framework of the paper. Based on the completed revisions, we re-examined the theme of the paper and revised it to "Intelligent Dance Motion Evaluation: An Evaluation Method Based on Key Frame Acquisition According to Musical Beat Features". The use of "multi-sensor alignment" is due to the constraint in this paper that requires audio and video data obtained from audio and video sensors to be aligned, which enables the acquisition of key frames in dance videos through musical beats. However, the main focus of this paper remains on action recognition and the development of an appropriate and comprehensive automated evaluation method. Based on this, we have made similar revisions.
- The entire method description section, which was originally divided into Chapters 2, 3, 4, and 5, has been integrated into new Chapters 2 and 3. Additionally, a detailed introduction to the entire method has been provided.
- We have significantly enhanced the implementation details and evaluation process of the evaluation method. In fact, it includes three levels of evaluation: evaluation of individual action features, which serves as the "element" and foundation of the evaluation; overall evaluation of all action features within a single action, which we have renamed as "Average Score based on the Cosine Similarity of Action Sequences (ASCS)"; and overall evaluation of action sequences consisting of a series of actions, which we have named as "Scoring Method Based on Action Contextual Relationships (SMACR)". This enables us to provide a comprehensive and accurate evaluation of the actions to be evaluated from different perspectives, such as teaching and training needs, and competition review needs.
Therefore, we believe that the original description of SMACR and SMACR+ was unreasonable, so we made significant revisions. In the newly revised Chapter 3, we reorganized the description of this part: we first introduced the similarity evaluation method for individual dance actions (in a single key frame), which is based on the evaluation method of key feature angle differences shown in Figure 7. In this way, all action features in a single key frame can form a feature sequence. Based on this feature sequence, a similarity evaluation method is constructed and compared with the standard feature sequence, thereby enhancing the overall value of action evaluation. This is also one of the important improvements compared with other methods in this paper, and we named it ASCS.
Furthermore, to consider the continuity of actions, we innovatively introduced the Rouge-L evaluation method from NLP, treating the actions corresponding to a single key frame as a word/token, and converting the evaluation of the entire action sequence into a similarity evaluation similar to that between sentences. This enables the entire action evaluation algorithm to evaluate from both the accuracy of individual actions and the coherence of actions, making the evaluation results more accurate and reasonable. This improved method is named SMACR.
Based on this adjustment, the main difference between ASCS and SMACR is that ASCS describes the accuracy (similarity) of actions within a single key frame, while SMACR describes the similarity (coherence) between key frame sequences, i.e., action sequences.
- While making significant revisions to the entire paper, we carefully revised the expressions in each part of the paper. We revised the English writing of the paper as much as possible and then applied for MDPI's English polishing service to improve the language organization level of this paper as much as possible. At the same time, some formatting errors in the paper were also revised.

Round 2
Reviewer 1 Report
Comments and Suggestions for Authors
In fact, authors improved the present manuscript. However, I have still detected several points that should be improved, mainly in the introduction and discussion parts.

Comments on the Quality of English LanguageThere are some paragraphs that an editing of English language is required.
Author Response
We sincerely express our gratitude to the review experts for their meticulous guidance and insightful comments, which directly addressed the core issues of our paper. After thoroughly studying the first round of review opinions, we carefully examined and adopted all valuable revision suggestions, making comprehensive and detailed improvements to the paper. Additionally, based on the supplementary literature, we conducted a new round of comparative experiments and verifications, striving to present our research findings and the contributions of this paper in a more adequate and comprehensive manner. Thank you again!
General comments
The authors should acknowledge the improvement made in the present manuscript. However, the introduction needs structural improvements (joint the item related work 1, with the introduction 0) for a better understanding of the question that will be answered by the authors. A discussion section should be included, once authors have a lot of data to confront with the specialized literature.
Taking into account the opinions of several other review experts, we first adjusted the main focus of the paper: one is to obtain key frames through musical beats, and the other is to select appropriate evaluation methods for action matching. Based on these revisions, we further revised the overall framework of the paper to elaborate on the implementation methods and details in a more detailed manner. We introduced comparative methods to verify the performance.
Abstract - In this section there is a huge discrepancy between background, objective, methodology, results and conclusion.
L1 to l17 This background is too long. No more than 5 lines to contextualize the theme.
L3to L5 Rewrite this sentence. Double words - dance movements
We have rewritten the background section and controlled the length of its description. We have also revised any repetitive expressions that existed within it.
L25 to L28 Is this sentence a result?
No. This is just the expected outcome of a standard "data preprocessing" process. That is, both the standard (action) data (videos) and the data to be evaluated should form a unified set of posture descriptions after this preprocessing. The difference lies in that the former serves as the evaluation benchmark, while the latter is the object to be evaluated. This representation corresponds to the "Standard Posture Descriptions Sequence" produced by the "Main Preprocessing" shown in Figure 1.
L27 Double word in the same sentence. Evaluation/dance evaluation.
L28 to L31 - Where are the results and conclusions?
Similar to Question 2, the expression used here may have caused misunderstanding. The formation of both the standard sample set (standard set) and the dataset to be evaluated (evaluation set) belongs to the preprocessing process. The ultimate goal, however, is to construct an appropriate evaluation method to assess actual actions. Therefore, we have adjusted this expression to focus on keyframe extraction and the evaluation conducted based on these keyframes.
Introduction
For a clean text and for the reader understand the question that authors want to respond,the authors should joint the introduction (0) with the related work (1).
We have enhanced the description in the introduction section to better enable readers to understand the research field and motivation of this paper. Combining with issue (x), we introduced the main work and contributions of this paper. On this basis, we have significantly strengthened the introduction of related work to fully present the development trends and existing problems in this field, leading to a solution proposed in this paper.
After comprehensive consideration, we have decided to retain separate sections for the introduction and related work.
L65-L77 This para authors is discribing about the methodology. Is this more convenient to insert in the methods section?
Yes. We have streamlined that part of the content and merged the detailed descriptions into the methodology section. Correspondingly, we have adjusted the framework of the paper to make the methodology introduction and implementation more compact.
L106-L109 Rewrite this sentence.
L136 Is this method section? Please, organize in accordance.
Here, adjustments have been made in conjunction with the methodology description.
L137-L145- A pair of references should be quoted here.
L153- L158 - A pair of references should be quoted here.
Relevant literature has been added. This corresponds to references 42 and 43 in the latest version of the paper.
Results
Figure 8 and Figure 9 These figures are with low resolution.
We have revised the presentation of Figures 8 and 9 to better showcase their key content.
Discussion
Why authors did not discuss the results presented? The section related work (1) could
help the authors to discuss all the data presented. Furthermore, there no discussion
about the methodology implemented, and none study limitations were presented.
In terms of dataset usage, popular datasets such as UCF101, HMDB-51, and Kinetics 400 are primarily applied in the field of action classification rather than action matching. Comparatively, datasets like KTH and Weizmann, while suitable for action matching tasks, are not ideal for direct evaluation of the method proposed in this paper due to the absence of audio files synchronized with the videos in their samples. Consequently, we reexamined existing research, particularly focusing on the ST-AMCNN method proposed in literature [44], which is representative in the field of action matching. Therefore, we selected ST-AMCNN as a comparison method, reanalyzed the existing data, and updated the test results.
Firstly, compared to using all video frames for action matching tests, selecting key frames based on musical beats significantly reduces the computational load for matching. Among the 12 test samples, an average of 2-5 key frames were selected every two seconds (corresponding to 2-5 beats), which is much lower than the 24 frames per second of the full frames, reducing the computational load to 4.1%-10.3% of the original (In other words, it was accelerated by 9.2 to 15.25 times.). It is noteworthy that when evaluating using only these key frames, the overall evaluation accuracy decreased by only 3%.
Secondly, in terms of matching accuracy for individual action frames, ST-AMCNN outperforms the ASCS method proposed in this paper. This is mainly because although the angle-based cosine matching method adopted by ASCS is simple, its matching accuracy significantly decreases when there are significant differences between the test subject and the standard sample, and the accuracy of multi-scale matching is also not high. However, the advantage of the ASCS method lies in its lack of need for pre-training and its smaller computational load.
Lastly, both ST-AMCNN and the ASCS method proposed in this paper can shift from evaluating individual action frames to evaluating action continuity with greater emphasis after incorporating the Rouge-L evaluation mechanism. Therefore, we present the fourth set of comparison test results here, namely ST-AMCNN*. It can be seen that, similar to the test results of SMACR, the improved evaluation method of ASCS, the score for continuity evaluation is lower than that for individual action frames, but the overall score is still higher than that of the SMACR method.
In summary, selecting appropriate key frames helps reduce the computational cost of subsequent matching methods, and depending on the specific application scenario, a faster evaluation method (such as SMACR in this paper) or a method with higher accuracy (such as ST-AMCNN) can be chosen.

Reviewer 3 Report
Comments and Suggestions for Authors
The more relevant works should be proper reviewed, e.g. Using Wearable and Structured Emotion-Sensing-Graphs for Assessment of Depressive Symptoms in Patients Undergoing Treatment, IEEE Sensors Journal, vol. 24, no. 3, pp. 3637-3648, Feb 2024. Wearable Structured Mental-Sensing-Graph Measurement, IEEE Transactions on Instrumentation and Measurement, vol. 72, pp. 1-12, Oct 2023.
Author Response
We sincerely express our gratitude to the review experts for their meticulous guidance and insightful comments, which directly addressed the core issues of our paper. After thoroughly studying the first round of review opinions, we carefully examined and adopted all valuable revision suggestions, making comprehensive and detailed improvements to the paper. Additionally, based on the supplementary literature, we conducted a new round of comparative experiments and verifications, striving to present our research findings and the contributions of this paper in a more adequate and comprehensive manner. Thank you again!
In addition to further enriching the literature, we have also supplemented a whole set of experiments based on the introduced literature to verify the role and value of keyframes.